# Relationship of mRNA Expression of Selected Genes in Peripheral Blood and Synovial Fluid in Cranial Cruciate Ligament Deficient Stifles of Dogs

**DOI:** 10.3390/ani12060754

**Published:** 2022-03-17

**Authors:** Karol Ševčík, Viera Karaffová, Marián Hluchý, Marieta Ševčíková, Zuzana Ševčíková, Valent Ledecký

**Affiliations:** 1Small Animal Clinic, University of Veterinary Medicine and Pharmacy in Košice, Komenského 73, 04181 Košice, Slovakia; sevcik.karol@gmail.com (K.Š.); marian.hluchy@uvlf.sk (M.H.); marietakurillova@gmail.com (M.Š.); valent.ledecky@uvlf.sk (V.L.); 2Department of Morphology Disciplines, University of Veterinary Medicine and Pharmacy in Košice, Komenského 73, 04181 Košice, Slovakia; zuzana.sevcikova@uvlf.sk

**Keywords:** cranial cruciate ligament, stifle, inflammation, osteoarthritis, extracellular matrix, cytokines, collagens, radiographic measurements, anatomic-mechanical angle

## Abstract

**Simple Summary:**

The cranial cruciate ligament rupture is characterized by chronic inflammation, osteoarthritis of the stifle joint, and extracellular matrix degeneration of the ligament itself in dogs. Early pre-clinical cranial cruciate ligament alteration cannot be detected by clinical examination or standard radiography. Therefore, we assessed the possible relationship of inflammatory markers in peripheral blood and synovial fluid of affected stifle joints in comparison to a control. We also evaluated components of the extracellular matrix of ruptured ligaments and finally compared the tibial plateau angle and the anatomical-mechanical angle between groups. Some of the assessed inflammatory markers were significantly increased in both the peripheral blood and synovial fluid compared with the control, as were collagens. The tibial plateau angle was not significantly different; however, the anatomical-mechanical angle significantly increased in the ruptured ligaments. Our results suggest a possible positive relationship between inflammatory markers of blood and synovial fluid in cranial cruciate ligament deficient stifles compared to the control. These findings may support both local and systemic inflammation process at the same time during osteoarthritis progression. Based on this, it would be interesting to investigate the predictive osteoarthritis pathway of inflammatory cytokines, *matrix metalloproteinases,* and their effect on the extracellular matrix components of the cranial cruciate ligament in future studies.

**Abstract:**

The cranial cruciate ligament rupture (CrCLR) is characterized by chronic inflammation and osteoarthritis (OA) of the stifle joint and extracellular matrix (ECM) degeneration of the ligament itself in dogs. Generally, OA may arise from chronic low-grade systemic inflammation. We assessed the possible relationship of inflammatory markers in the peripheral blood (PB) and synovial fluid (SF) of affected stifle joints in comparison to a control. Moreover, no study has shown the possible association between PB and SF levels of inflammatory markers in CrCLR stifles of dogs in veterinary medicine yet. We also evaluated components of ECM of CrCLR and finally compared the tibial plateau angle (TPA) and the anatomical-mechanical angle (AMA) between groups. Samples from PB and SF were examined for mRNA expression of interleukins, TNF-α and INF-γ. ECM components—collagen 1A1 and 3A1 and elastin—were examined for mRNA expression from SF. The level of relative expression for IL-1β, IL-8 and IFN-γ was significantly increased in both PB and SF in CrCLR stifles as compared with the control. Collagens were also significantly increased in CrCLR stifles. TPA was not significantly different; however, the AMA angle significantly increased in the CrCLR group. Our results suggest a possible relationship between PB and SF levels of inflammatory markers in CrCLR stifles of dogs.

## 1. Introduction

The exact etiopathogenesis of a cranial cruciate ligament (CrCL) rupture remains unknown despite extensive research. The degenerative changes occurring in the CrCL, it does not matter whether the source of degeneration is biological or biomechanical [1], are responsible for a progressive reduction of its elasticity and mechanical resistance, which make it more susceptible to even minimal trauma. This situation may promote degeneration and lead to partial or total rupture, with the coexistence of progressive osteoarthritis (OA) and inflammation of the stifle joint [2,3].

The ligament itself, synovium, and articular cartilage are described as tissues with inflammatory changes. Synovitis, the most discussed possible biological source of CrCL degeneration, is typically characterized by the expression of proinflammatory mediators and recruitment of inflammatory cells [4]. Multiple cytokines, such as interleukin (IL)-1b, IL-6, IL-8, IL-10, IL-17 and tumor necrosis factor-alpha (TNF-a), are known to be involved in stifle inflammation due to CrCL disease [4,5,6,7,8]. The development of synovitis can lead to a significant reduction in the tensile strength of the CrCL. As is known, the structural properties of the cruciate ligament depend on the composition of the extracellular matrix (ECM) in addition to inflammatory influences. ECM turnover is a normal physiological process in which synthesis is balanced by degradation, but any imbalance may promote degeneration and lead to ligament rupture [9]. The extracellular matrix of a ruptured CrCL has increased matrix turnover compared with an intact CrCL [10]. However, whether the increased matrix turnover forms part of the rupture mechanism or part of a reparative response after CrCL rupture is not yet known [11]. Although collagen provides tensile strength to the ligament, other structural components, such as elastin, most likely contribute to the overall mechanical function of the complex [12] and recently have been shown to form up 20% of the CrCL [13]. The exact role of elastic fibers in the CrCL is still unknown but they may be involved in the provision or maintenance of elasticity, stabilization of blood vessels, anchoring tissue, or guidance of cell migration [14,15]. It was recently stated that the number of elastic fibers, which may be part of the reparative fibers, is increasing with the degree of ECM degradation in intact stifles of Labrador Retrievers and Greyhounds [14,16].

Biomechanical factors, such as stifle joint morphology including morphological deformities of the proximal tibia and distal femur, malalignment, altered kinematics, or distorted joint contact areas may play a role in CrCL degeneration [10,17,18,19,20,21,22,23]. Numerous studies have been studying the tibial plateau angle (TPA) extensively. However, the true effect of TPA on CrCL stresses in vivo is still not clear [24,25]. Parameters other than TPA have been suggested to be more relevant in conjunction with CrCL rupture. Malalignment between the anatomical and mechanical axes of the tibia may be induced as a consequence of the caudal angulation of the proximal tibia, which may result in the proximal anatomic axis not fully aligned with the longitudinal anatomic axis [26]. The angle between the anatomical and mechanical axes of the tibia (AMA angle) is used to quantify caudal angulation of the proximal tibia [27,28,29], and it has recently been stated that AMA angle magnitude may be a clinically relevant predisposing factor for the development of CrCL rupture in dogs [30].

Early pre-clinical CrCL changes in the stifle joint of dogs cannot be detected by clinical examination or standard radiography. Therefore, the assessment of molecular changes in the peripheral blood (PB) and synovial fluid (SF) might be an area of interest in CrCL, or generally in OA-related research [31,32]. It has recently been indicated that changes in PB levels of pro-inflammatory cytokines could influence levels found in the SF [33]. To the best of our knowledge, no study has shown the possible association of systemic-peripheral blood and local-synovial fluid inflammation in CrCL deficient stifles of dogs in veterinary medicine. Therefore, we decided to evaluate relative gene expression of selected genes in stifle joints of dogs with CrCL rupture and unaffected control stifle joints. Consistent with this part, we hypothesized that there might be a simultaneous increase in PB and SF levels of selected inflammatory markers in our cohort of patients. 

Our second aim was to evaluate inter- and intra-observer measurements of TPA as traditional radiographic measurement associated with stifle morphology and compare it with AMA angle as a possible new more precise marker of CrCL risk factor. 

## 2. Materials and Methods

### 2.1. Animals

All client-owned dogs included in this study were presented to the Small Animals Clinic of the University of Veterinary Medicine and Pharmacy in Košice (Slovakia), between May 2019 and April 2021. All owners of the dogs used in this study agreed with the collection of data, and all dogs underwent clinical examination performed according to a standardized orthopedic protocol [34].

Dogs included in the CrCL group (*n* = 50) had naturally occurring, unilateral, surgically confirmed, partial or complete rupture of the CrCL, and no evidence of any other concurrent stifle pathology upon physical and radiographic examinations. Additional inclusion criteria were: being free of systemic disease based on a complete physical; neurologic and orthopedic examinations; lack of significant abnormalities on a complete blood count and serum biochemistry profile; and not receiving any medication for at least one month prior to sampling.

Inclusion criteria for the control group (*n* = 15) were conditional on the absence of any history of an orthopedic disease (particularly stifle joint) and not receiving any medication for at least one month prior to sampling. Subjects with known malignant hematological or oncologic conditions, chronic infection, or inflammation were excluded from this study. All dogs were euthanatized by intravenous administration of barbiturates for reasons unrelated to this study.

### 2.2. Sample Collection and Isolation of Total RNA

#### 2.2.1. Blood

Before surgery or euthanasia, blood samples were collected to RNeasy Protect Animal Blood tubes (Qiagen, Manchester, UK) and stored at 25 °C for 2 h and then kept at −20 °C prior to total RNA purification. RNA purification was carried out by an RNeasy Protect Animal Blood Kit (Qiagen, Manchester, UK) according to the manufacturer’s protocol. Both the purity and concentration of isolated total RNA were measured spectrophotometrically on an Implen nano-photometer (Implen, Munchen, Germany) at ratios of 260/280 and 260/230. For all RNA samples, the ratio was 260/280 around 2.0 and the 260/230 ratio was around 1.8 to 2.2, indicating that protein contamination was minimal. Then, 1 μg of the total RNA was immediately reverse transcribed with an iScript cDNA Synthesis Kit (Bio-Rad, Hercules, CA, USA).

#### 2.2.2. Synovial Fluid

Immediately before the stab incision for the arthrotomy during surgery, synovial fluid (SF) was aspirated from the affected stifle joints. SF was obtained by percutaneous arthrocentesis from the stifle joints of control dogs immediately after they were euthanatized. The intactness of the stifle joint and the CrCL was confirmed radiographically and also by arthrotomy. Samples with visible blood contamination were excluded. RNA purification from SF was carried out by an RNeasy mini kit (Qiagen, Manchester, UK) in accordance with the manufacturer’s protocol. Both the purity and concentration of isolated total RNA were measured spectrophotometrically on an Implen nano-photometer (Implen, Munchen, Germany) at ratios 260/280 and 260/230. For all RNA samples, the ratio was 260/280 around 2.0 and the 260/230 ratio was around 1.8 to 2.2, indicating that protein contamination was minimal. Then 1 μg of the total RNA was immediately reverse transcribed using oligo-DT- primers and iScript cDNA Synthesis Kit (Bio-Rad, Hercules, California, USA) and subsequently diluted with nuclease-free water to the total volume of 30 µL. 

The protocol for reverse transcription to cDNA was described in Karaffová et al. [35].

### 2.3. Relative Expression of Target Genes in Quantitative Real-Time PCR (qRT-PCR)

The mRNA levels of cytokines (IL-1β, IL-8, IL-10), TNF-α, INF-γ in PB and SF as well as collagen 1A1 (COL1A1), collagen 3A1 (COL3A1), and elastin (ELN) in SF were determined. A total of 5 reference genes (GAPDH, TBP, RPL13A, HPRT, RPS5) were tested in triplicate on three plates. Finally, mRNA relative expression of the most stable reference gene encoding glyceraldehyde-3-phosphate dehydrogenase (GAPDH) was determined based on the stability of expression using geNorm software (Table 1). The primer sequences used for qRT-PCR are listed in Table 2. All primer sets allowed DNA amplification efficiencies between 94% and 100%.

Amplification and detection of specific products were performed using the CFX 96 RT system (Bio-Rad, USA) with Maxima SYBR Green qPCR Master Mix (Thermo Scientific, Waltham, MA, USA). The reaction mix (25 µL) consisted of: 2.5 µL of cDNA template, 2× Sybr green master mix (12.5 µL) and 10 µL mix of specific primers (forward and reverse primer) for each gene. The qRT-PCR for detection of the relative gene expression was initiated by denaturation at 94 °C for 3 min and was followed by 39 cycles consisting of denaturation at 93 °C for 45 s, annealing at 60 °C for 30 s, and was finalized by an elongation step at 72 °C for 10 min. A melting curve from 50 °C to 95 °C with readings at every 0.5 °C was produced for each individual qRT-PCR 96-well plate. Analysis was performed after each run to ensure a single amplified product for each reaction. All real-time PCR reactions were performed in duplicates, and mean values of the duplicates were used for subsequent analysis. We also confirmed that the amplification efficiency of each target gene (including GAPDH) was between 94% and 100% in the exponential phase of the reaction, where the quantification cycle (Cq) was calculated. The Cq values of the genes studied were normalized to the average Cq value of the reference gene (^Δ^Cq), and the relative expression of each gene was calculated mathematically as 2^−ΔCq^.

### 2.4. Radiographic Procedure and Measurements

Radiographs were obtained using a standard clinical X-ray unit (Gierth HF 200A, X-ray apparatus, GIERTH GmbH, Riesa, Germany), digitized with a computed radiography system (FCR Prima T2, CR-IR 392, Computed Radiography, Fujifilm Co., Tokyo, Japan) and saved as DICOM files. All radiographs were carried out on sedated or euthanized dogs. Sedation was achieved by butorphanol (Butomidor, Richter Farma AG, Wels, Austria) (0.2 mg/kg) and medetomidine (Cepetor, CP-Pharma MbH, Burgdorf, Germany) (15 μg/kg). Stifle and hock joints were kept at approximately 90°, in accordance with previous criteria [42].

TPA was measured as previously described [42]. The AMA angle was measured as the angle created by the anatomical and the mechanical axis of the tibia. (Appendix A) [26]. The tibial anatomic axis was defined as a line connecting the midpoint between the cranial and caudal cortex of the tibia at 50% and 75% of the tibial shaft length [29,43]. The mechanical axis was defined as a straight line connecting the midpoint between the intercondylar tubercles of the tibial plateau with the center of the talus [26,44].

All radiographic measurements were repeated two times with a two-month interval, by two observers, who were appropriately trained and familiar with measurements. For each of the two assessments, radiographs were presented in a random order to help ensure on the second occasion that the observer was unaware of their initial angle.

### 2.5. Statistical Analysis

Data were tested for normality using the Shapiro—Wilk test. Animal data and almost all cytokine and collagen mRNA expressions appeared to be normally distributed; therefore, were expressed as mean (M) and standard deviations (SD). While AMA values were not normally distributed, all TPA values were normally distributed, so both radiographic measurements were interpreted as mean ± SD and median with interquartile distance (IQR).

A parametric t-test was used to compare age and body weight between dogs with partially and completely ruptured ligaments. For mRNA expressions of selected genes, the parametric t-test and Wilcoxon-Mann-Whitney rank-sum test were used to compare parametric and non-parametric data between the CrCL group and the control group, and between partial and complete CrCL rupture, respectively.

Intra- and inter-observer agreement was evaluated with the two-way random single measures intra-class correlation coefficient for absolute agreement (ICC 2,1) [45]. Measurements were grouped separately for inter-observer agreement, and for intra-observer, they were grouped as the first and second round. The ICC ranged from 0 (no agreement) to 1 (perfect agreement). ICC < 0.5 was considered as poor reliability, values between 0.5 and 0.75 indicated moderate reliability, values between 0.75 and 0.9 indicated good reliability, and values greater than 0.90 indicated excellent reliability [46].

Differences were considered significant at *p* < 0.05. All statistical analysis was performed using IBM SPSS version 27 statistical software.

## 3. Results

### 3.1. Animals

Data on body weight, gender, reproductive status, the state of rupture (partial or complete), and the presence of meniscal damage were collected (Table 3). Several breeds were included in the CrCL group: 8 American Staffordshire Terriers, 7 Labrador Retrievers, 5 Golden Retrievers, 5 Rottweilers, 4 Boxers, 4 Doberman Pinschers, 3 Akita Inu, 3 Central Asia Shepherd Dogs, 3 German Shepherd Dogs, 3 mixed-breed dogs, 2 *English Bulldogs*, 2 *Hungarian Vizslas*, 1 Belgian Shepherd Dog. The control group included: 7 mixed-breed dogs, 4 Dachshunds, 2 Greyhounds, 2 Whippets.

Twenty-seven of 50 (54%) dogs had a complete rupture, and 17 (63%) of them had medial meniscus tear (MMT). The other 23 (46%) dogs had a partial rupture, of which 5 (21.7%) had a medial meniscus tear. No significant difference was found in age (*p* = 0.91) or body weight (*p* = 0.27) between dogs with complete and partial CrCL rupture.

### 3.2. The Relative Expression of Selected Genes

#### 3.2.1. Blood

The levels of relative expression for IL-1β (*p* < 0.001), IL-8 (*p* < 0.001), TNF- α (*p* < 0.01) as well as IFN-γ (*p* < 0.001) were markedly increased in CrCL group compared to the control (Figure 1a–d). On the other hand, IL-10 gene expression was not significantly different between groups (*p* > 0.05) (Figure 1e).

#### 3.2.2. Synovial Fluid

Likewise, the levels of IL-1β (*p* < 0.0001), IL-8 (*p* < 0.001) and IFN-γ (*p* < 0.001) gene expression in SF were significantly higher in the CrCL group than in the control (Figure 1f,g,i). On the other hand, IL-10 (*p* < 0.01) and TNF-α (*p* < 0.05) gene expressions were significantly downregulated in the CrCL group in comparison to the control (Figure 1h,j). Both collagen gene expressions were significantly increased in the CrCL group compared to the control group (*p* < 0.001) (Figure 2a,b). The level of ELN gene expression was increased in the control, as compared to the CrCL group (*p* < 0.001) (Figure 2c).

### 3.3. Comparison of mRNA Expressions of Cytokines, Collagens and Elastin between Partial and Complete Ruptured Ligaments

No significant differences were found between expressions of any cytokine, collagen or elastin between partial and complete CrCL rupture in either PB or SF (*p* > 0.05) (Table 4.).

### 3.4. Radiographic Measurements

All results are shown in Table 5. The CrCL group and the control group differed in AMA angle significantly (*p* = 0.004), while the dogs with complete and partial rupture did not (*p* = 0.29). No significant difference was found in TPA between either the CrCL and the control groups (*p* = 0.95), or between complete and partial rupture (*p* = 0.35).

To assess the inter-observer reliability of measurements, the agreement between two observers using the same methods was determined. ICC was excellent for AMA-0.91 and moderate for TPA-0.88.

To assess the intra-observer reliability of measurements, the agreement between the two repeated measurements of each observer was made. ICC for AMA for the first observer was 0.94 and for the second was 0.93, which means it was excellent for both. For TPA, it was 0.88 for the first observer and 0.94 for the second, meaning moderate for the first observer and excellent for the second.

## 4. Discussion

It was demonstrated that changes that take place in the synovial lining, in the ligament itself, and in the cartilage are also initiated by pro-inflammatory cytokines, which are subsequently dispersed by synovial fluid [47,48]. The possible direct relationship between PB and SF inflammatory markers in human medicine with knee OA has recently been suggested [33]. In veterinary medicine, such a possible relationship has not yet been described. Therefore, this study tries to address this issue besides achieving other goals.

IL-1β is considered to be one of the major players involved in OA pathophysiology. The level of IL-1β is elevated in the synovial fluid, synovial membrane, subchondral bone, and cartilage [49], and more recently it was detected also in the PB of patients with OA in human medicine [33] and infrapatellar fat pad (IFP) of CrCL deficient dogs [50]. Until now, IL-1β has not been evaluated from PB in patients with CrCL disease. In our findings, the level of IL-1β was significantly higher in stifle joints with CrCL rupture than in the control group in both PB and SF. Moreover, this cytokine induces the production of several other cytokines and chemokines involved in OA progression, including IL-8 [51]. IL-8 has been found to have a major role in angiogenesis, one of the key mechanisms involved in the maintenance of chronic synovial inflammation [52]. In the presented study, a significant difference in the upregulation of IL-8 expression was found between stifle joints with CrCL rupture and the control group in both PB and SF. SF observations are in accordance with previously reported findings [7,53] and, additionally, Garner et al. [54] suggested IL-8 as an SF candidate biomarker of stifle OA. Our values of INF-γ were significantly elevated in the patient group compared to the control in PB and SF as well. Blood results are in line with those previously reported in human patients with OA of the knees, where the authors proposed that INF-γ may serve a crucial role in the development of OA [55]. On the other hand, Page et al. [56] suggested a need to rethink the doctrine of the pro-inflammatory function of IFN-γ in autoimmune diseases, such as rheumatoid arthritis (RA), because of its ability to reduce IL-1β expression in the arthritic joint. Whether the same situation exists in CrCL disease is not clear yet, but the potential effect of therapeutically induced INF-γ could be beneficial. Based on the knowledge of the authors, none of these three markers was determined from the blood of CrCL deficient stifles. These findings may suggest a possible relationship between PB and SF presence of these cytokines, which may be beneficial in early diagnostics of OA changes due to CrCL disease.

Following the Starling–Landis theory [57], which states that the fluid exchange between plasma and interstitial space is driven by the equilibrium of hydrostatic and oncotic pressure across the capillary wall, it was further suggested that changes in blood levels of pro-inflammatory cytokines could influence levels found in the synovial joint. In keeping with this theory, the authors speculate that in cases where all other factors influencing the levels of IL-1β, IL-8, and INF γ in PB can be excluded, including Borreliosis-associated arthritis, the levels of these cytokines in PB can be further tested for a direct relationship with the SF levels in CrCL deficient stifles. In the case of positive findings of pro-inflammatory markers between PB and SF, this could help in the early diagnosis of CrCL disease as blood sampling is much easier compared to SF. Moreover, this may open a debate on another hypothesis of the initial CrCL alteration based on the Starling-Landis theory and the transmission of systemic inflammatory markers to the stifle joint and the alteration of its homeostasis. Additionally, a very recent model of the OA pathway suggested that increased IL-8 expression in PB may be associated with an increase of IL-18 and matrix metalloproteinase-3 (MMP-3) expression in the knee joint [33]. This model does not suggest a direct relationship between the same marker in PB and SF, but the influence of one marker in PB on another marker in SF. It would be interesting to verify this in the context of CrCL disease in dogs.

TNF-*α* is one of the mediators secreted in early osteoarthritis [58]. IL-1*β* and TNF-*α* drive the inflammatory cascade independently or in collaboration with other cytokines [59]. They induce the production of a number of other cytokines [51,60] and MMPs [61] that further alter the joint environment. In the presented study, TNF-α expression was significantly higher in PB of CrCL deficient stifles in comparison to the control group but was significantly downregulated in SF of patients. These results could be consistent with findings in humans, where it was reported that TNF-α levels were increasing in serum and decreasing in SF as the stage of OA progressed, and the values were reversed at a lower stage of the disease. These data show a locally increased effect of TNF-*α* in the early stages, whereas in the advanced stages, it is increased systemically [62]. These findings can be extrapolated to ours, as the CrCL rupture is the final stage of progressive disease. Upregulation in serum and downregulation in SF suggests responsibility for systemic inflammation with a decreasing direct effect on the joint. Upregulation in serum is probably the source of the production of other pro-inflammatory markers. More recently, a significant increase of TNF-α in IFP but not in SF has been published [50]. A possible explanation is the potential of adipose tissue to act as a source of inflammatory factors. Furthermore, it has an intra-articular but extrasynovial structure in the knee and stifle joint [63].

Our COL I values were significantly higher in CrCL deficient stifles compared with the control group. These findings are in line with previous reports [10,64]. In addition, CrCL ligamentocytes subjected to higher uniaxial tensile cyclic loading have been reported to increase COL I expression [65]. Contrary to our results, Ichinohe et al. [66] described reduced COL I levels in artificially created canine models with high TPA compared to contralateral limbs that were free of TPA alteration. A possible explanation for these differences is that the CrCLs in the study of Ichinohe et al. were not ruptured, although chondroid metaplasia and ECM alteration were observed. This may be explained by the findings of Zachos et al. [67] who described the alteration of caudal cruciate ligament morphology after CrCL transection as a consequence of acute changes in repetitive tensile force, which is possibly consistent with the acute change of TPA in the Ichinohe study. Additionally, Beagles, the breed used in the Ichinohe study, are not typical dogs with a predisposition to CrCL rupture; therefore, the processes in the ligament may be slightly different. Significantly increased COL 3 expression was observed in this study, which concurs with previously described findings [64,66]. COL 3 has traditionally been thought of as repair collagen, and the increase in its expression suggests that the CrCL is indeed attempting, but is unable, to bridge the gap as described in the reparative phase [68]. Previously, it has been described that a breed predisposed to CrCL rupture has an increased number of ELN fibers as degeneration advances. However, there were lower numbers of these fibers compared with the non-predisposed breed. These findings may indicate the healing potential of the ligament, as a greater amount of oxytalan fibers are commonly seen in the healing response [16]. On the other hand, these authors also hypothesized that with increasing grade of degeneration, these changes are true indicators of disease. Our results showed the opposite trend—the amount of ELN was significantly lower in the CrCL deficient stifles compared to the control group. It is difficult to compare these results with ours because their samples were from intact stifles, whereas ours were from CrCL deficient stifles. A potential explanation is that a higher number of ELN fibers have been localized in the healing areas of arteries [69], myocardium [70], and muscles [71], which are highly vascular tissues compared to the CrCL, which is nearly avascular. In addition, the healing capabilities of ELN may be altered in ruptured ligament and chronic inflammation of the entire joint. As previously mentioned, both IL-1β and TNF-α stimulate the production of pro-inflammatory cytokines and MMPs through the activation of several signaling pathways [72], which are associated with increased ECM degeneration [4]. Therefore, the authors of this study hypothesized that it would be interesting to evaluate the direct relationship of several cytokines and MMPs between PB and SF as possible markers of CrCL ECM alterations, especially collagen.

True effect of TPA on CrCL in vivo is still a controversial topic. On the other hand, the AMA angle has recently been described with promising results as a potential clinically relevant risk factor for CrCL deficiency [30,73]. Our median values of AMA angle in CrCL ruptured stifles were significantly higher in comparison with the control group. Both inter- and intra-observer agreements were excellent for the AMA measurements, compared to TPA where inter-observer and one intra-observer agreements were good and the second observer agreement was excellent according to previously described criteria [46]. These results are consistent with previous data suggesting that the AMA angle is a more accurate predictor of CrCL disease than TPA [30,73]. Furthermore, we achieved higher observer agreement in AMA angle compared with TPA.

## 5. Conclusions

To the best of our knowledge, a simultaneous evaluation of inflammatory markers in PB and SF in CrCL deficient stifles has not been published to date. Our results suggest a possible positive relationship between inflammatory markers of PB and SF in CrCL deficient stifles compared to the control group. At the same time, it would be appropriate in the future to focus on monitoring other genes that may be involved in the inflammatory process using other methods such as RNA seq.

These findings may support both local and systemic inflammation processes at the same time during OA progression. This relationship may not only be related to the direct influence of one marker among PB and SF, but an increase of one marker in PB may affect the level of another marker in SF. Based on this, it would be interesting to investigate the predictive OA pathway of inflammatory cytokines, MMPs, and their effect on the ECM components of CrCL in future studies. Diagnosis of OA markers from the peripheral blood based on predictive pathways could help in the early diagnosis of not only CrCL alteration, but possibly also OA of other joints in dogs as well. Moreover, only hypothetically, following the Starling–Landis theory [57], systematic anti-inflammatory cytokine treatment modalities targeting the joint might be effective for early CrCL tears of stable stifles or as protection of contralateral stifle. We also agree with the statement that AMA angle could be a potential clinically relevant factor for CrCL deficiency.

## Figures and Tables

**Figure 1 animals-12-00754-f001:**
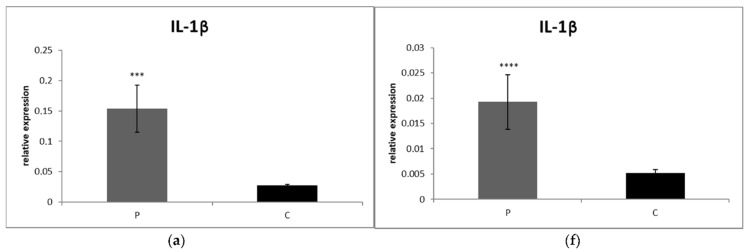
Level of relative gene expression for (**a**,**f**) IL-1β, (**b**,**g**) IL-8, (**c**,**h**) TNF-α, (**d**,**i**) IFN-γ (**e**,**j**) IL-10. Results at each time point are the median of 2^–ΔCq^. Columns marked with stars are significantly different between groups: * *p* < 0.05, ** *p* < 0.01, *** *p* < 0.001, **** *p* < 0.0001. P, patients CrCL group; C, control group.

**Figure 2 animals-12-00754-f002:**
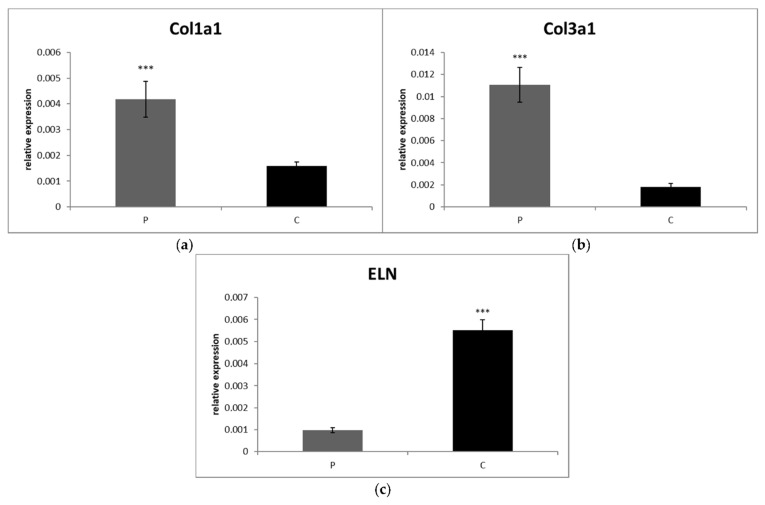
Level of relative gene expression for (**a**) Col1a1, (**b**) Col3a1 and (**c**) ELN (SF). Results at each Table 2. ^Cq^. Columns marked with stars are significantly different between groups: *** *p* < 0.001. P, patients CrCL group; C, control group.

**Table 1 animals-12-00754-t001:** Ranking of reference genes according to their expression stability by GeNorm.

Genes	Stability Measure M value
GAPDH	0.333
RPL13A	0.404
TBP	0.495

Legend: GAPDH, glyceraldehyde-3-phosphate dehydrogenase; RPL13A, ribosomal protein L13A; TBP, TATA, box binding protein.

**Table 2 animals-12-00754-t002:** List of primers used in qRT-PCR for target genes mRNA of dog.

Primer	Sequence 5′–3′	References
IL 1β Fw	CTGTGTGATGAAGGATGGAA	[36]
IL-1β Rev	AATCGCTTTTCCATCTTCCT	
IL-8 Fw	GAACCGCAATCCTACTTTTG	
IL-8 Rev	GATCATTCAACCCAGCATTG	
IL-10 Fw	TGCATGGCTCAGCACTGCTCTGTTG	[37]
IL-10 Rev	AGTGGGTGCAGTCGTCCTCAAGTAG	
TNF-α Fw	TCTCGAACCCCAAGTGACAAG	[36]
TNF-α Rev	CAACCCATCTGACGGCACTA	
IFN-γ Fw	AAGATCAGCTGAGTCCTTTG	
IFN-γ Rev	AAATCACGCAAAGCTGAAAA	
Col1a1 Fw	AGAGCATGACCGACGGATTC	[38]
Col1a1 Rev	ACGCTGTTCTTGCAGTGGTA	
Col1a3 Fw	CTGAAGGAAACAGCAAATTC	[39]
Col1a3 Rev	ATTCCCCAGTGTGTTTAGTG	
ELN Fw	GGCCTGGGAATTGGTGGTAA	[40]
ELN Rev	CTCTTCCGGCCACAGGATTT	
GAPDH Fw	CATGTTTGTGATGGGCGTGAA	[41]
GAPDH Rev	GATGACTTTGGCTAGAGGAGC	

**Table 3 animals-12-00754-t003:** Study population characteristics. MMT, medial meniscal tear; RS, reproductive status (the number indicates castrated/spayed dogs); BW, body weight; M, male; F, female.

Group	No. of Stifles	MMT	Sex (M/F)	RS (M/F)	Age (y)	BW (kg)
CrCL R	50	22	22/28	12/18	5.9 ± 2.3	32.3 ± 10.8
Complete	27	17	13/14	8/10	6 ± 2	30.5 ± 11
Partial	23	5	9/14	7/5	5.9 ± 2.9	35.5 ± 10.1
Control	15	-	9/8	-	8.2 ± 3.8	22 ± 12.8

**Table 4 animals-12-00754-t004:** Comparison of mRNA expressions of cytokines, collagens, and elastin between partially and completely ruptured ligaments. Numbers indicate the *p* value. No significant difference was found.

Parameter	PB	SF
IL 10	0.30	0.21
IL-1b	0.38	0.46
IL-8	0.06	0.43
TNF-α	0.48	0.42
IFN-γ	0.49	0.26
Col 1	-	0.18
Col 3	-	0.22
ELN	-	0.09

**Table 5 animals-12-00754-t005:** Mean values, standard deviations (SD), median, and interquartile range (IQR) of radiographic measurements of AMA and TPA. Values are expressed in degrees.

Group	AMA		TPA	
	Mean ± SD	Median; IQR	Mean ± SD	Median; IQR
CrCL R	2.99 ± 1.03	2.9; 1.17	24.96 ± 2.84	25.45; 3.93
Complete	3.02 ± 1.20	2.55; 1.48	25.01 ± 3.24	25.69; 4.35
Partial	2.8 ± 0.6	2.95; 1.02	25.23 ± 2.49	25; 4.26
Control	1.81 ± 0.29	1.94; 0.45	24.91 ± 1.15	25.05; 1.77

## Data Availability

The data presented in this study are available on request from the corresponding author.

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
