# Peer review of "Relationship of mRNA Expression of Selected Genes in Peripheral Blood and Synovial Fluid in Cranial Cruciate Ligament Deficient Stifles of Dogs"

_animals, 2022, doi:10.3390/ani12060754_

Round 1

Reviewer 1 Report

I think it is an article that addresses a topic of interest in the use of diagnostic biomarkers for osteoarthritis, specifically taking the anterior cruciate ligament injury as a model.
I recommend its publication due to the originality of the work and the contribution of new data that can serve as a reference for future works related to diagnosis, follow-up and response to treatment in osteoarthritis.

Author Response

Dear reviewer,

Many thanks for your support and opinion.

Reviewer 2 Report

This is well prepared scientific report which concerned a simultaneous evaluation of inflammatory markers in peripheral blood and synovial
fluid in CrCL deficient stifles. The subject of the article is very original and innovative. In my opinion, the article can be published in it is current form.

Author Response

(The authors gave the same response as above.)

Reviewer 3 Report

In this manuscript, Sevcik et al studied the mRNA expression of selected genes in peripheral blood and synovial fluid in cranial cruciate ligament deficient stifles of dogs. This may be a valuable finding in the field of veterinary inflammation and no doubt this is a very well-thought out project. Hence, it may be nice addition to MDPI-Animals. However, the authors concluded their finding following a gene set that I think is too limited. I realize that these are good representative genes, but still sort of cherry-picked. The authors’ conclusion is presumably right based on these genes, but there could be many other genes that may be involved in this inflammation process. To include a whole gene signature, the authors should do RNA-seq comparing samples. During present days, this is not a difficult assay to perform that is necessary to see the whole picture including the ontology of differentially expressed genes, enrichment of pathways in comparison to publicly available datasets, and most importantly, select candidate genes for future functional studies. Given this I think this is a decent study but it could be definitely better by including RNA-seq that would make this study from mediocre to a highly ranked publication. If other reviews are favorable, I would at least like the authors to comment on this point in conclusion section.

Author Response

Thank you for your valuable comments. Regarding RNA-seq, I consider it important to mention the fact that our experiment was pilot in this theme and therefore we have so far used only a few known genes to monitor this interaction, which could be associated with CrCLR stifles of dogs in veterinary medicine. In truth, I don't work with RNA-seq and I don’t have experiences with it, but in the future we will certainly do this methodology and complete the whole picture, including the ontology of differentially expressed genes. Definitely, I agreed with you that our study could be better if RNA sequencing was added. I added comment about this point in conclusion section (marked with yellow colour).

Round 2

Reviewer 3 Report

I thank the authors for considering my point and adding the relevant sentence in the conclusion section - “At the same time, it would be appropriate in the future to focus on monitoring other genes that may be involved in the inflammatory process using other methods such as RNA seq.” I think this very nicely sets up a logical platform for future subsequent studies that can follow this manuscript, and at the same time, takes care of the current limitation of the study at this point. Given the other reviews, I think this manuscript can be accepted in this current revised form. Congratulations to all the authors.

Author Response

Dear Editor,

Thank you for your support once again.